# Interventions for Child Drowning Reduction in the Indian Sundarbans: Perspectives from the Ground

**DOI:** 10.3390/children7120291

**Published:** 2020-12-14

**Authors:** Medhavi Gupta, Sujoy Roy, Ranjan Panda, Pompy Konwar, Jagnoor Jagnoor

**Affiliations:** 1The George Institute for Global Health Australia, University of New South Wales, Level 5, 1 King St, Newtown, NSW 2042, Australia; mgupta@georgeinstitute.org.au; 2Child in Need Institute, Daulatpur, Pailan, South 24 Parganas, West Bengal 700104, India; sujoy@cinindia.org (S.R.); ranjan@cinindia.org (R.P.); 3Injury Division, The George Institute for Global Health India, 311-312, Third Floor, Elegance Tower, Plot No. 8, Jasola District Centre, New Delhi 110025, India; pkonwar@georgeinstitute.org.in

**Keywords:** drowning, child health, injury, low-and middle-income country, India, preventative medicine, implementation science, qualitative research

## Abstract

Drowning is a leading cause of child death in the coastal Sundarbans region of India due to the presence of open water, lack of supervision and poor infrastructure, but no prevention programs are currently implemented. The World Health Organization has identified interventions that may prevent child drowning in rural low-and middle-income country contexts, including the provision of home-based barriers, supervised childcare, swim and rescue training and first responder training. Child health programs should consider the local context and identify barriers for implementation. To ensure the sustainability of any drowning prevention programs implemented, we conducted a qualitative study to identify the considerations for the implementation of these interventions, and to understand how existing government programs could be leveraged. We also identified key stakeholders for involvement. We found that contextual factors such as geography, cultural beliefs around drowning, as well as skillsets of local people, would influence program delivery. Government programs such as accredited social health activists (ASHAs) and self-help groups could be leveraged for program implementation, while Anganwadi centres would require additional support due to poor resourcing. Gaining government permissions to change Anganwadi processes to provide childcare services may be challenging. The results showed that adapting drowning programs to the Sundarbans context presents unique challenges and program customisation.

## 1. Introduction

Drowning is a leading cause of morbidity and mortality in low-and middle-income countries (LMICs) [1]. Of these deaths, 62,000 occur in India, where drowning is the foremost cause of death by injury for children aged 1–4 years [2]. Rural and remote coastal regions in LMICs present the highest risk of child drowning. Rural, forested Sundarbans region in the northern state of West Bengal is one such area. Sundarbans experiences frequent flooding, a presence of open water, poor infrastructure and poor health systems [3,4,5]. A recent survey in the Sundarbans found particularly high rates of drowning in children aged 1–9 years where it is likely the leading cause of death in this group [6].

The World Health Organization (WHO, Geneva, Switzerland) recommends the implementation of four effective community-based interventions in rural LMIC settings to reduce drowning in young children. These interventions are activities that may be feasibly implemented in low-resource contexts and have been shown to reduce drowning burden [1]. These interventions are: the installation of home-based barriers controlling access to water (such as playpens and door barriers), the provision of supervised safe spaces with capable child care, teaching school-aged children basic swimming and rescue skills and training adult bystanders in rescue and resuscitation [1]. Previous research has shown that communities in the Sundarbans consider drowning a health issue [7]. Despite this perception and the high rates of drowning, there are no preventive measures implemented in the region.

Previous research and experience in the sustainable program design has shown that it is essential to understand the context, local perceptions and possible implementation-related challenges before designing and implementing community-based programs [8,9,10]. The identification of key stakeholders that may support or inhibit implementation must also be identified [11,12]. These stakeholders can include members of the community who can influence program engagement, as well as governmental or organisational leaders whose support and buy in is beneficial for community acceptance and access to local resources.

A key strategy that improves program sustainability is linking program goals with government priorities and leveraging existing programs [13,14]. A comprehensive policy review of West Bengal and National policy found three government programs that may be appropriate to build upon to implement drowning reduction programs: the integrated child development scheme (ICDS), self-help group (SHGs) schemes and the accredited social health activist (ASHA) program [15]. The federal ICDS program was introduced in 1975 and aims to provide free childcare services to children aged 3–6 years through village-based Anganwadi centres [16]. The implementation and reach of these centres are highly variable across the Sundarbans, and many centres do not provide the childcare services promised in the policy (Biswas and Chattapadhyay, 2001; Biswas et al., 2010). The quality improvement of the ICDS program has the potential to provide structured supervision, for injury prevention. The SHG scheme aims to reduce rural poverty and increase household income through the setup of self-help groups in villages, primarily with women. Some SHGs also become involved in community projects, such as the provision of midday meals in schools [17]. ASHA workers are community-based health workers who focus on child and maternal health on an incentive-based system, and have close ties with mothers [18,19]. Both SHGs and ASHAs may be leveraged in the provision of community-education such as rescue and resuscitation training and supporting families in building and maintaining home-based barriers.

We conducted the formative contextual analysis required to design a sustainable drowning reduction program for the Sundarbans, as guided by WHO recommendations. The objectives were as follows: (1) identify community perceptions and preferences towards the recommended drowning interventions; (2) explore the feasibility of leveraging ICDS, ASHA or SHG programs for the delivery of drowning reduction interventions; (3) identify contextual challenges and considerations for the design and delivery of the program; and (4) identify key stakeholders who should be engaged during the development and implementation of the program.

## 2. Materials and Methods

We applied qualitative methods to understand the micro context in which drowning reduction interventions could be delivered in the Sundarbans. In-depth interviews (IDIs), focus group discussions (FGDs) and observations were conducted and triangulated to develop this understanding. IDIs gave insights into individual-level perspectives, and FGDs were used to identify community norms and perceptions. Observations allowed for the better understanding of government program operations and systems. Qualitative methodology was guided by the Consolidated Criteria for Reporting Qualitative Research (COREQ) (see Appendix A) [20].

### 2.1. Data Collection

Data collection was conducted in partnership with a local non-governmental organization (NGO), the Child in Need Institute (CINI). CINI has operated child and maternal health programs in rural West Bengal for the past 46 years and has extensive connections with communities and local government in the Sundarbans region. The data collection of community-based participants was completed by two male data collectors recruited by CINI, managed by S.R. and R.P. who work as the programs’ manager and director, respectively. The data collectors had previous experience in qualitative research in West Bengal and were trained by the researchers in the study aims and tools. One of the data collectors had experience conducting qualitative data collection in the Sundarbans and was familiar with the community. M.G. conducted English-language interviews, such as with grassroots organisations.

All data collection occurred face-to-face. IDIs and FGDs were held in locations that best suited participants, such as in community schools or Anganwadi centres. In addition to the data collectors and participants, NGO partner facilitators were present for some IDIs, FGDs and observations to lend logistical support. All IDIs and FGDs were audio recorded and lasted between 30 and 90 min. Field notes were also taken by one data collector and collated to make key point summaries of each IDI, FGD and observation on a daily basis, which was shared with the research team.

All Bengali transcripts were translated into English for analysis. No interviews were repeated. Transcripts were not returned to participants for comment due to the logistical and literacy barriers.

### 2.2. Participants

All participants were adults over the age of 18. A minimum of three IDIs and two FGDs were conducted for each stakeholder type to ensure the capture of varying responses. Stakeholder types interviewed included (1) community-level participants including men, women and leaders, (2) government program participants and delivery staff, and (3) grassroots organisations. These stakeholder types are described in detail below. This range of stakeholders enabled for the identification of contextual considerations from the perspectives of program implementers and beneficiaries and ensured that individuals from different levels of the social hierarchy were heard. Data collection ceased once saturation in each type was reached.

#### 2.2.1. Community-Level Participants

Community men, women and leader participants were recruited through convenience sampling. The partner NGO first approached local government bodies for permission to conduct the interviews. Data collectors then entered the communities as recommended by the Gram Panchayats (who are the lowest local government body representing a group of villages) and engaged local leaders such as ASHA and Anganwadi workers, who introduced the data collectors to possible participants. Participants were required to be parents and usual residents of the community living there for the past three years. Participants were recruited across all 19 blocks of the Sundarbans to ensure a range of perspectives.

These participants provided insights into community acceptance and perceptions towards drowning interventions, as well as possible barriers and enablers to implementation in the local context. A total of ten IDIs and nine FGDs were conducted with community-level participants, with men and women equally represented.

#### 2.2.2. Government Program Participants

Anganwadi workers, SHG members and ASHA workers were approached through purposive sampling after entering communities in which permission for data collection had been granted by local government officials. Communities had one or two ASHA and Anganwadi workers each, so whoever was available upon contact was scheduled for interview. As self-help group members were found in many households, community members would lead data collectors to the closest home of a member. These participants were included if they were active in their respective programs for a minimum of 6 months within the Sundarbans region.

Observations of SHG meetings and Anganwadi centres were conducted to understand their operations. By policy, one Anganwadi centre is required to serve a population of 1000 people, providing any children aged 3–6 years old with early childhood education activities for two hours each day along with a nutritious meal. Each centre should have one Anganwadi worker and one helper, and usually operates between 7 and 10 a.m. We observed the children who came to the Anganwadi Centres and provided insights into how children interacted with the Anganwadi workers, as well as the ground realities of the program delivery. Observations of SHGs identified decision-making methods and revealed the role of SHGs in the community. Anganwadi centres and SHGs for observations were purposively selected in partnership with a local NGO working with these programmes to cover a range of performance levels. Nine government program participants (ASHA workers, Anganwadi workers and SHG members) were interviewed. Two FGDs with SHG members were also held and three observations each were conducted at Anganwadi centres and SHG meetings.

#### 2.2.3. Grassroots Organisations

Interviews were conducted with the individuals from organisations working in the child health, education, safety or nutrition in the Sundarbans or other similar rural contexts in West Bengal. This provided insights into the considerations and challenges related to delivering grassroots programs in the Sundarbans. Potential participants were introduced to the researchers by our partner NGO and were required to have oversight over program delivery for at least one year. Three representatives from grassroots organisations were interviewed.

### 2.3. Tools and Transcriptions

Tools for all IDIs, FGDs and Observations were developed before the commencement of data collection and translated into Bengali. The data collection guides were semi- structured to ensure all domains relevant to research questions were covered.

All participants were also shown a pictorial presenting the WHO-recommended drowning interventions. Barriers were described as any physical object preventing children’s access to water such as playpens, door barriers or fencing. Childcare was described as any group-based supervision in an enclosed space. Swimming lessons encompassed both swim and rescue training skills, and first responder training was described as training adults on how to save children if they fall into water or start drowning.

### 2.4. Ethics

All participants provided verbal or written informed consent depending on their literacy level. Ethical approval was granted by the University of New South Wales Human Research Ethics Committee (HC 190274) in Sydney, Australia and The George Institute for Global Health (India) Ethics Committee (06/2019) in New Dealhi, India.

### 2.5. Analysis

Analysis of the transcripts was completed using NVivo 12 [21]. Narrative analysis was used where key themes under each of the broad research objectives were derived. All transcripts were coded against a priori key themes based on the research questions, including the acceptability of each of the WHO drowning reduction programs, and considerations for the implementation and feasibility of using government programs to deliver the programs. Subsequent sub-themes were developed under each of these based on commonalities and diversified perspectives from participants. We also triangulated different sources of data by coding for the type of stakeholder and type of qualitative method (IDI, FGD, observation) to assess congruent and different perspectives across genders and participant type as well as to compare individual and community-level viewpoints [22]. The two independent reviewers (M.G. and P.K.) discussed their results and discrepancies before finalising the key findings.

Stakeholders were identified and then allocated to level of power and interest as based on Mendelow’s Matrix [23]. The level of power describes the stakeholder’s influence over program success, and the level of interest reflects the impacts of the program on the stakeholder. The framework was used to identify the correct engagement strategies for each of the stakeholders based on their framework allocation.

## 3. Results

Refusal to participate in the study was less than 10%. Below we discuss the overall and intervention-specific considerations for program implementation in the Sundarbans, the feasibility of using government programs and identify the stakeholders who must be involved in program design and delivery.

The Appendix A (Appendix A in Appendix A) depicts illustrative quotes from the following analysis. Figure 1 below provides a summary of the main enablers and barriers identified for the intervention implementation, from the perspective of program beneficiaries (demand-side) and from the perspective of program implementers (supply-side).

### 3.1. Considerations for Program Design and Delivery

A range of considerations were identified that applied to all drowning reduction interventions.

#### 3.1.1. Acceptability across All Interventions

Participants showed heterogeneity in preferences between the interventions. All interventions were generally considered acceptable. Some participants recognised that each of the interventions targeted different age groups and expressed a need for an age-targeted and comprehensive approach (Refs 1 and 2 in Appendix A of Appendix A).

#### 3.1.2. Affordability

Cost to households was a concern for all interventions. Many participants stated that with limited resources and competing priorities, a drowning reduction intervention would not be affordable for households. The home-based barriers’ intervention was considered the most feasible for self-funding as it was viewed as a one-time investment, with maintenance being of negligible cost. They also noted that parents who are unable to afford services may cause problems and complain if excluded. Some participants suggested that families could pay different amounts depending on their income level, which could be pooled together to fund the program (Ref 3 in Appendix A).

#### 3.1.3. Community Engagement and Ownership

Consistent community engagement through regular meetings, showcases, theatre and household visits were identified as important to implementation success. Participants noted that program ownership should be transferred to the community over time, such as by setting up an implementation committee. Participants noted that without consistent engagement, people may fall back into previous habits and stop engaging with the program (Refs 4 and 5 in Appendix A).

Community leaders and grassroots organisations’ participants also discussed the importance of regular program monitoring. They stressed that communities and implementing agencies should work in partnership to ensure that interventions were being implemented and used as designed (Ref 6 in Appendix A).

#### 3.1.4. Resources and Skill Set

Participants also noted that geographical and infrastructure barriers such as the road quality and the connectivity of many areas were challenges. Participants suggested that local resources should be used where possible, such as bamboo from the area for barriers (Ref 7 in Appendix A).

Grassroots organisations also noted that finding capable human resources was often challenging due to lower educational attainment in the region and the migration of skilled workers to the cities. Benefits and incentives would need to meet community expectations to recruit capable staff. However, the programs would provide an opportunity for women to access employment, as few jobs were available to them post high-school. Program providers may also face risks if a child was injured under their care from angry parents (Ref 8 in Appendix A).

#### 3.1.5. Social Class

Participants largely stated that caste and religion did not present an issue. Community member participants did not anticipate any discrimination towards potential intervention beneficiaries. However, some instances of discrimination against Muslim Anganwadi workers by Hindus, or against Hindu SHG members by Muslims, were reported by government program participants during IDIs. Government program participants also stated that political party affiliation may affect cooperation and participation in interventions. Program staff from different political parties may refuse to work together or may discriminate against communities from other parties (Refs 9 and 10 in Appendix A).

Some participants also noted that as Muslims were relatively economically disadvantaged and conservative, they may have less capacity or willingness to pay (Ref 11 in Appendix A).

### 3.2. Intervention-Specific Considerations

In addition to findings to guide general program implementation, specific considerations were identified for each of the WHO-recommended drowning interventions.

#### 3.2.1. Home-Based Barriers

Home-based barriers were largely acceptable to communities provided certain conditions were met. Many adults noted that this method was used previously in their childhood but concerns for children’s mental wellbeing stopped the practice as a lack of movement and social interaction with other children was considered detrimental. The intervention was also only considered suitable for younger children under the age of 2–2.5 years, as older children would try to climb the barriers (Ref 12 in Appendix A).

The feasibility of different types of barriers varied between households. Some participants noted that families may struggle to keep door barriers and pond fencing gates closed due to regular access. Building and maintaining fencing around all ponds within 20–50 m of homes may not be feasible due to the large number of ponds in some villages. Some community members expressed concerns over restricting children’s movement in playpens which may be detrimental to their development (Refs 13 and 14 in Appendix A).

For playpens, many participants noted that an adult would still need to be present to ensure safety. Participants also suggested that door barriers or fencing gates could be made lower so that adults could climb over without opening them, increasing convenience and reducing the likelihood of it being left open. One participant suggested that young children from nearby homes could be kept together in a large playpen in the middle of the homes with one adult supervising (Refs 15 and 16 in Appendix A).

Some participants identified that locally trained professionals were required to build and install the barriers to maintain quality (Ref 17 in Appendix A).

#### 3.2.2. Childcare and Supervision-Based Programs

Childcare was largely acceptable in communities, especially as it provided parents with relief from supervision while they worked and offered an opportunity for children to participate in early childhood education, including for children with disabilities who often had few avenues for learning (Ref 18 in Appendix A).

However, some participants were concerned for children’s safety, as one adult was not considered enough supervision for a group. The region had also experienced instances of child trafficking. Parents were also busy during the day and often restricted in their ability to pick up and drop off children. This issue would be exacerbated in monsoon season when roads are flooded. Parents were also concerned that young children below the age of two years old would not engage with activities and experience separation anxiety.

Participants offered a range of suggestions for childcare implementation. Children could be divided into groups by age so they could be engaged in age-appropriate activities (Refs 19 and 20 in Appendix A).

Participants stated that more than one carer was required to look after children to ensure they remained supervised if one child had to be taken for a bathroom break. They also supported the employment of a trusted and known local woman with training for the role (Ref 21 in Appendix A).

The provision of toys, activities and learning material was also required to ensure that parents and children would be interested. A gated ‘community playground’ was suggested to provide an outdoor play space. Pick up and drop off services would increase attendance. Toilet and water facilities were also required. Food provision would improve attendance as both parents and children would be more satisfied. The venue was also required to be large and secure for safe play (Refs 22 and 23 in Appendix A).

The preferred hours for the childcare services varied. Many participants, especially mothers, noted that parents were busy in both the morning and afternoon but were home for lunch. They suggested a session both before and after lunch (Ref 24 in Appendix A).

#### 3.2.3. Swim and Rescue Training

Many participants believed that children had adequate swimming skills from informal lessons provided by parents in family ponds but were interested in rescue training. Participants acknowledged that some individuals did not have access to a pond to learn or did not have time to teach their children swimming, so classes were important for them. Children would also be motivated by the chance to participate in regional swimming competitions (Ref 25 in Appendix A).

Ponds in this region are mostly privately owned and used for washing, cleaning and fishing. Some participants reported that there were no common ponds large enough for training in their communities, so private ponds were required. Seeking someone who would lend their pond may be difficult. In addition, many ponds were unsuitable, being dirty and deep (Ref 26 in Appendix A).

For quality control, participants suggested that guidelines for pond selection should be developed, covering location, cleanliness and depth criteria. Safety and rescue material should also be available, and platforms built for access to the pond. A changing room would also reduce community push-back as children would not travel home in wet clothing (Refs 27 and 28 in Appendix A).

Some participants believed a trainer from outside the community would be better respected by the community, while others preferred a local who would have better relationships with the community and would be more consistently available (Refs 29 and 30 in Appendix A).

#### 3.2.4. First Responder Program

The first responder training was acceptable to most participants, especially parents of young children as they were interested in learning how to protect them (Ref 31 in Appendix A).

However, cultural beliefs may remain a barrier to appropriate responses. During drowning events, people in the communities had previously ignored health advice from community health workers such as ASHAs and conducted traditional responses, such as calling a local village doctor or performing rituals on the water. These responses had led to delays in children receiving appropriate medical care (Ref 32 in Appendix A).

#### 3.2.5. Indigenous Interventions for Child Safety

A range of other intervention ideas and solutions were offered by participants. Many stated that awareness programs were required in parallel to drowning interventions to educate communities about the risks of drowning and ensure sustained behaviour change. Awareness activities would also seek to dispel harmful beliefs about drowning, such as on cultural post-drowning rituals that led to delays in children receiving first aid. Participants noted that other existing programs in communities with established activities, such as vaccination programs, could be leveraged for awareness activities (Ref 33 in Appendix A).

Native interventions employed by communities were also identified. Some parents tied their children to their waist or to the house with rope while they worked. Others kept their children locked inside the home alone when they were away (Ref 34 in Appendix A).

Other possible solutions were offered such as providing vans for school children or organising ‘walking buses’ where children would travel to school together, and teaching children to have a ‘shore guard’ during play time where one child kept watch from the pond’s edge.

### 3.3. Use of Government Programs in Drowning Intervention Delivery

Possible roles in the implementation of drowning interventions were identified for existing government programs in communities.

#### 3.3.1. ASHA Workers

ASHA workers were interested in supporting the dissemination of drowning reduction programs and were considered suitable for providing training due to their reputation as health workers. ASHA workers were already regularly visiting mothers and children up to the age of 5 years old and could encourage the use of drowning interventions and conduct checks of home-based barriers. However, some participants noted that not all ASHA workers had strong relationships with communities, where their health communications such as community meetings were now largely ignored due to fatigue with repeated advice and instructions (Ref 35 in Appendix A).

ASHAs already had some skills in rescue and response. Some ASHAs expressed a desire to learn first aid more comprehensively to perform better in their roles. They were also willing to train others in their communities. However, ASHA workers stated that their work was highly unpredictable as they often responded to calls of women in labour, and so could not provide training and childcare for large blocks of time (Ref 36 in Appendix A).

ASHA workers worked on an incentive-based system and expected added payment for services. (Ref 37 in Appendix A).

#### 3.3.2. Self-Help Groups

Self-help group (SHG) members were primarily interested in the delivery of childcare services and suggested they may provide pick up and drop off services for children to and from swimming and childcare interventions. Some SHGs were already involved in delivering government programs, such as the mid-day meal scheme in schools. They expected to be paid for involvement (Ref 38 in Appendix A).

Many households in communities had at least one SHG member, making them well connected. They would be able to support community engagement activities, such as through organising mothers’ meetings and household visits (Ref 39 in Appendix A).

Some possible barriers for the engagement of SHGs were identified. Firstly, many were busy with their family businesses and may have minimal time to be engaged. Secondly, some were concerned about their lower levels of education and stressed the need for comprehensive training. Lastly, some SHGs others faced challenges with the engagement of all members. The supervision of SHGs also varied and the management of SHGs involved in drowning intervention delivery may require a separate system (Ref 40 in Appendix A).

Community leaders and grassroots organisation participants noted that SHG members were easier to engage in drowning interventions than Anganwadi centres or ASHAs as they required fewer government permissions. However, some SHG members may face restriction from their husbands or families due to cultural constraints on women’s mobility and employment (Ref 41 in Appendix A).

#### 3.3.3. Anganwadi Centres (ICDS Program)

Anganwadi centres were considered possibly suitable for the implementation of childcare supervision and parent engagement activities. Centres were usually open from 7 a.m. to 9 or 10 a.m. with 20–30 children attending each day. Some centres already provided a limited range of childcare activities, and parents left their children for 1–2 h with the Anganwadi centre.

However, there was great variability described and observed in the quality of services. Participants reported that many Anganwadi centres only provided food and no childcare services. This may be due to the lack of an appropriately enclosed venue, lack of training for Anganwadi workers and parents’ low trust in the centre. In two out of three of the Anganwadi centre observations, the Anganwadi worker did not facilitate any games or activities. Many participants also complained of a lack of educational materials and repeating activities (Refs 42 and 43 in Appendix A).

In addition, many venues lacked toilets and water and children were left alone if a child was taken to relieve themselves. Many participants reported that Anganwadi venues did not have enough space for both cooking and childcare activities and were not safely enclosed. A barrier to finding appropriate venues was that the local government requested private land to be leased for 50–100 years for the centres, which few people agreed to. Parents also did not always have time to pick and drop their children, especially if the centre was at a further distance from their home (Ref 44 in Appendix A).

Anganwadi workers were also burdened with their duties and had limited training. Anganwadi workers had other responsibilities such as conducting surveys for the Department of Health on sanitation and maintaining the registers of children. They were often busy until 12 p.m. after the centre closed at 10 a.m. They also struggled to cook, clean and provide childcare activities at once. Many centres did not have an Anganwadi assistant allocated or regularly attending. Anganwadi workers also reported being unsatisfied with the pay (Refs 45 and 46 in Appendix A).

Anganwadi workers were trained when they joined the program, but the training did not cover ECE activities in detail. They were provided limited ongoing support, where meetings with Panchayat officials who oversaw the implementation of ICDS, visits from supervisors and block-level offices were infrequent (Ref 47 in Appendix A).

Parents also complained that food was of inadequate quantity. Improper food provision meant many parents had lost trust in the Anganwadi centres. Anganwadi workers and community leaders stated that poor food quality was due to resourcing issues such as insufficient money provided for ingredients amidst rising prices and a lack of water and sanitation in the venues (Ref 48 in Appendix A).

Making changes to Anganwadi centres at a local level required permissions from both Health and Women and Child Development representatives at the block level. Block-level representatives (the level of government just above Gram Panchayats) are responsible for monitoring program performance. Although Gram Panchayats are responsible for the program implementation of ICDS, they do not have the permission to make operational changes as their targets and delivery requirements are set by State policy and enforced by block-level supervisors. Grassroots organisation and community leader participants noted that engaging block-level representatives may be challenging without higher state-level permissions which may take months to obtain. Grassroot participants had experienced that government departments were cautious about giving permissions when liabilities were not clear. These participants stated that running a parallel program for childcare may be easier than using the ICDS (Refs 49 and 50 in Appendix A).

A few communities had parallel NGO-run childcare programs which children attended after visiting the Anganwadi centre. These programs were considered of better quality than Anganwadi centres (Ref 51 in Appendix A).

#### 3.3.4. Other Community Programs

Local youth clubs were identified by many participants as potential implementers of programs. These clubs were organisations run by youth and overseen by Gram Panchayats. They aimed to engage young people in self and community development activities (Ref 52 in Appendix A).

### 3.4. Stakeholder Analysis

A range of stakeholders important to program delivery were identified. The placement of these stakeholders along Mendelow’s Matrix is presented in Figure 2 below.

#### 3.4.1. Block-Level Officials

Block-level government officials represent the district-level government at a smaller administration level. Information and gaining permissions from the block-level government would ensure there were no complaints later down the line, especially if existing government programs were used (Ref 53 in Appendix A).

#### 3.4.2. Gram Panchayat

The Gram Panchayat was the lowest local government body. They were responsible for the ICDS and SHG programs. They ran awareness and door-to-door campaigns on issues such as dengue. Most participants stated that any implementation activities must involve the Gram Panchayat. The Gram Panchayat would give permission for activities occurring in communities and were influential over village leaders. They could also assist in the recruitment of suitable program staff, the identification of venues for intervention activities and assist in resolving arising challenges. However, Gram Panchayats were unlikely to have their own funding to support the interventions (Ref 54 in Appendix A).

Issues with nepotism in the Gram Panchayat operations were reported. Positions and resources were given to family members to run schemes who had limited incentive to ensure quality. This may present a challenge with recruitment. Panchayats were also not always responsive to requests for resources. One Anganwadi worker had been submitting applications for a new venue for two years with no response (Ref 55 in Appendix A).

#### 3.4.3. Community Leaders

Many communities had a leader or influential educated individuals. These individuals would need to be engaged before implementation to assist with delivery and community mobilisation. This may include the village head and teachers. They also oversaw the activities of SHGs (Ref 56 in Appendix A).

#### 3.4.4. Local Police Stations

Local police stations would become involved if any accidents or issues occurred, so participants suggested that they should be made aware of any intervention activities. This would ensure they were willing to assist if challenges arise (Ref 57 in Appendix A).

#### 3.4.5. Community Members

Community members should take an active role in the implementation of the drowning interventions, providing inputs in locations and responsibility. Community-level participants were interested in supporting the programs (Ref 58 in Appendix A).

#### 3.4.6. Engagement Strategies

As per the placement on Mendelow’s Matrix, community members, village leaders and Gram Panchayats had high levels of power over program implementation and a high level of interest. Hence, these stakeholders should be actively and directly engaged in intervention design, development and implementation. Block-level officials have a high level of power, but lower levels of interest and should be engaged for permission and kept informed.

## 4. Discussion

Our analysis of micro and community-level stakeholder perceptions towards drowning interventions revealed opportunities for the implementation of drowning reduction programs in the Sundarbans.

The findings suggested that all recommended interventions must be introduced together in a comprehensive program for maximum effectiveness. According to the participants, barrier-based interventions were considered appropriate for 1–2-year-old children, childcare for 3–5-year-old children, and swim and rescue training for children over the age of 6 years. Participants were largely homogenous in this view, given cultural norms around childrearing and care. In addition, first responder training was perceived as important to encourage appropriate post-drowning actions. The age-appropriateness identified by participants for each intervention was in line with WHO implementation guidelines [24].

While the core components of these interventions would remain the same, such as ensuring that childcare spaces are secure and are provided during at-risk hours, the delivery processes of a comprehensive program should be adapted to the Sundarbans context. These changeable program characteristics include the nature of the community delivery agents, the capacity and capability of available workforce, availability of infrastructure and resources, partnership opportunities, methods of communication, and cultural adaptations such as changes to language and messaging [25,26]. The design and development of the comprehensive drowning program should involve community groups and stakeholders to ensure sustainability.

In this study, some specific intervention adaptations were identified as appropriate to the Sundarbans. An essential finding for the barrier-based intervention was that the preferred type of barrier varied by household. Hence, a drowning program may seek to deliver customised barriers for each household. Participants identified that childcare services should have an adequate child to caretaker ratio to ensure child security and provide pick up and drop off services to encourage attendance. These provisions to ensure child safety and support for attendance were also identified in international guidelines on childcare provision [27,28]. Participants were similarly concerned with safety for swim and rescue training services.

Participants also noted the need for complementary awareness activities, such as to dispel improper beliefs around effective child rescue techniques. Common responses to child drowning incidents involve engaging local quack doctors to perform rituals and trying to remove water by spinning the child over an adult’s head [7,29]. Changing problematic norms and beliefs is an important step in behaviour change, and Sundarbans communities must be informed that such actions do not save children [30,31,32,33]. However, awareness itself is not sufficient to change behaviour, and must be accompanied with the removal of obstacles to change and capacity building [31,34]. Hence, awareness and first responder training in the Sundarbans may also need to target local ‘quack’ doctors who have some authority over community responses to drowning and may override individuals advocating for the administration of proper first aid. Ensuring that these local doctors themselves promote and administer appropriate first response may be critical for sustainable impact.

The sustainability of programs improves when they leverage existing government structures [9]. Our findings suggest that the ICDS, ASHA and SHG programs may provide platforms through which a drowning reduction program may be promoted and implemented. ASHA workers may play a promotive and monitoring role for the program and may also be involved in first responder training. However, many ASHA workers are overburdened with their duties and their drowning program role may be more sustainable if it is incorporated into their existing activities, such as providing barrier monitoring support as part of their regular household visits [35]. SHG members also showed willingness to be involved with drowning reduction activities and provided a network through which program activities can be advertised. Members were also available to be recruited for program delivery. ASHA worker and SHG members’ performance may also vary depending on the frequency of visits from government supervisors, so independent program monitoring may be required [36].

Concerns were raised around the utilisation of Anganwadi centres for childcare services. The ICDS program suffered from unsafe venues, lack of Anganwadi training and poor sanitary conditions. The local government also had limited authority over the changing operations of centres to include more hours of childcare, requiring permissions from state-level bureaucrats, which may take time given decision makers are risk-averse to changes. NGO participants suggested that a parallel program was more feasible. The long-term goal of health program design, implementation and scale up is often the uptake of these programs by government, as this improves the likelihood of sustained funding and delivery [14,37]. A parallel program may be less likely to be picked up by the government as the ICDS program already provides childcare services as per policy. In addition, optimising existing Anganwadi centres may require fewer resources than opening new centres. Community and local-government engagement activities should seek to decide on which model has long-term feasibility: optimising Anganwadi centres or running a parallel program.

Key facilitating factors that will enable implementation were identified by participants. Consistent community engagement and buy in of local leaders were essential. This is well founded in other LMIC contexts [38]. However, participants also noted that local government was affected by nepotistic practices that may affect program quality. In West Bengal, a study found that local government members were allocating agricultural resources to communities with more power, land and connections [39]. Hence, strict protocols and oversight may be required to ensure the equitable distribution of program resources.

Community participants advocated for local individuals to be trained as childcare and swim training providers. Implementation analyses have shown that local community-based workers best operate when they have access to resources, training and monitoring. Additionally, the building of soft skills, such as communication and leadership, is vital [40]. Community worker engagement and management should be carefully defined and involve incentive structures appropriate to the context and matching community expectations [41,42,43].

Participants also noted that community-level committees are effective mechanisms through which residents can own programs and monitor implementation. These committees can also be engaged in advocacy and engagement activities and be instrumental in ensuring that implementation responds to community needs [26]. Increased community ownership of health programs may lead to better adaptation to the context and a greater likelihood of sustainability and acceptability [12,44]. However, the underlying assumption of all participants was that an NGO with expertise in child programs, such as CINI, would take primary lead in implementing and supporting community-level committees and program delivery. CINI has over 46 years of experience in delivering child programs in rural regions of West Bengal and is a suitable lead agency.

The development of the intervention may also consider the incorporation of other ideas. Although there is limited evidence on the effectiveness (and on the ethics) of tying children indoors to the ends of rope, there is some evidence that walking school bus programs can prevent injury in children [45]. While these have previously been used to reduce road traffic injuries, in the context of the Sundarbans, this may help reduce drowning events during commutes to school [46]. However, this intervention does not target the age group with the largest burden—1–4-year-old children.

To ensure the community ownership and development of an acceptable and feasible program, the next step of program design should involve community participatory approaches [10,47]. The present study found a range of issues that may affect program delivery, such as unpredictable geography, poor connectivity, religion and the caste of program providers, remoteness, poverty, poor government program monitoring structures, requirement for appropriate incentives, recruitment challenges and the availability of appropriate venues. Communities are the best informants for how context-specific issues can be addressed and managed [48]. Participatory approaches will also improve community buy-in and redistribute the power of change into the community’s hands [49]. The range of stakeholders as identified in the stakeholder analysis and should be appropriately engaged, starting with Gram Panchayat and block-level officials and moving to individual community leaders and members. No lifesaving organisations were identified which conduct drowning prevention activities, which was unsurprising as lifesaving organisations have had limited contribution to drowning prevention capacity development and advocacy in remote regions of India.

### Limitations of This Study

Due to ethical constraints, we were not able to gather information on participants’ caste or religion. It is unclear if the perspectives found are representative across a range of religious groups. In addition, some government program workers were recruited with assistance from Gram Panchayats. These may have been the more active and well performing workers and may not be fully representative of typical programs. This was particularly mitigated by ensuring at least one poor performing worker of each type was purposively recruited.

## 5. Conclusions

The Sundarbans are a high-risk region for child drowning, and we aimed to identify the mechanisms and considerations for the implementation of drowning reduction programs in the region. We found that program design should consider contextual factors such as geography, cultural beliefs around drowning, skillsets of local people and household-level needs. It was found feasible to leverage government programs such as ASHA workers and SHGs for program recruitment and implementation, while the optimisation of Anganwadi centres for the provision of childcare may be challenging due to poor resourcing and permissions required. Community-based young clubs were also possible implementers of programs. Program development and implementation should involve a range of stakeholders such as local government members, block and district-level health and development officials, community leaders and residents. The results show that the development of drowning reduction programs in rural LMIC contexts should be catered to the local social and environmental context to ensure acceptability and feasibility.

## Figures and Tables

**Figure 1 children-07-00291-f001:**
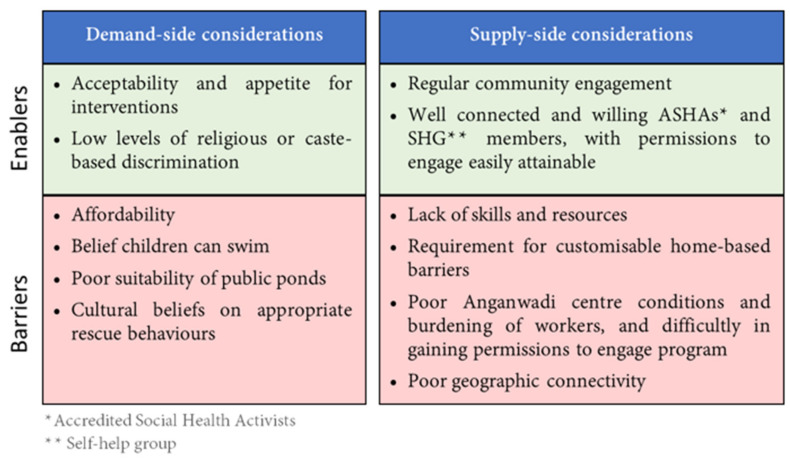
Key contextual enablers and barriers to implementation identified by participants.

**Figure 2 children-07-00291-f002:**
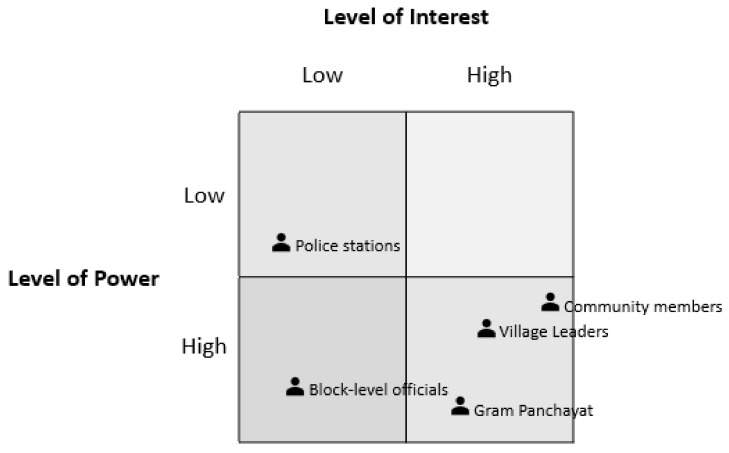
Stakeholder placement in Mendelow’s Matrix.

## Data Availability

Due to the qualitative nature of data collection, some participants may be identifiable from the contextual factors presented in the transcript. Data will be shared by the corresponding author on reasonable request.

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
