# Peer review of "Interventions for Child Drowning Reduction in the Indian Sundarbans: Perspectives from the Ground"

_children, 2020, doi:10.3390/children7120291_

Round 1
Reviewer 1 Report
A good paper. Only suggestion is that there should be no use of acronyms--e.g.in the Abstract.
Reviewer 2 Report
General
This paper addresses an important area the barriers and opportunities surrounding implementation of normative guidance. Objectives were described as:
- Identify community perceptions
- Explore feasibility of linking a number of existing programs
- Identify barriers to implementation
- Identify key stakeholders
It may be the journal preference, but I found the discussion suggested tables would be physically at a different place than in line with text. In addition, tables were not available in the manuscript reviewed. I would suggest that tables appear in line so one is not moving around within the paper.
The paper would benefit from a more detailed but succinct description of each of the following prior to discussion:
- Interventions
- Stakeholder groups
- Governmental relations
Specific
Line 45: WHO does not recommend pond fencing as it is generally impractical. Barriers WHO does recommend include playpens, swimming pool fencing and door barriers.
107: A clearer description of stakeholder type should be provided.
122: Snowballing may need some further detailed explanation of how it was done and whether any steps were taken to minimize bias.
168: Refusal to participate was very low and in context of snowballing above needs to be discussed. If generally everyone approached wishes to share views, engage in FGD etc etc then presumably one could have employed a sampling framework to ensure one elicited truly representative views of the communities rather than snowballing.
183: What were type of barriers discussed in the home-based barriers segment? More broadly, the authors have not to this point clearly delineated what were the main constituent features of all of the interventions being discussed with community members.
229 on: Here barriers are discussed in more detail but prior to this point a clearer description of the interventions should have been provided.
270: What is meant by swim training from “home lessons”?
316: Some further explanation of why ASHA messaging had become ignored would seem instructive since it suggests an existing program underperforms. Understanding why would seem relevant to the thrust of this study.
360: A clearer description of the Anganwadi would be helpful – how many personnel, what types of activities did they do, ages of children accepted etc. If this info was provided earlier in a brief descriptive segment in the methods it would make interpretation of the results segment easier.
375: The governmental control/approval issues briefly touched on would presumably be an issue for expansion of the Anganwadi centres to cater to drowning prevention needs. Little exploration of this. Unpacking in particular the relationship between Gram Panchayat and the higher levels of government upon which they might depend would seem important and this discussion could then be framed as something like “local government authority” rather than GP so as to make the paper more broadly applicable.
380: tpyo – that should be than
435: childcare for children from the age of mobility onwards (e.g. 12 months) is important. Unclear whether finding of daycare being suitable only for those 3-5 is opinions of participants or if investigators explored feasibility of daycare for children under 3 and found it was not deemed feasible. The discussion here sounds rather mutually exclusive for programmatic approaches and these interventions should not be seen as “either/or”.
Reviewer 3 Report
This study aimed to ensure drowning program sustainability for the Sundarbans in India by conducting a qualitative study to identify considerations for the implementation of WHO recommended interventions, and to understand how existing government programs could be leveraged. The formative contextual analysis required to design a sustainable drowning reduction program was guided by the WHO recommendations. Such studies are widely recognised as necessary to inform successful interventions.
A complex series of qualitative studies was undertaken with various sectors of the relevant community. A schema would be useful in the paper to summarise these studies and to indicate how they inter-relate. Similarly, greater use of tables/figures would likely simplify presentation of some of the results – possibly in a cross-cutting matrix format.
Would the national life saving association in India (associated with the International Life Saving Federation) be able to assist with local drowning prevention programs? Were they consulted in the development of the study?
Was consideration given in the study to a suitable lead agency to oversee drowning prevention program implementation, co-ordination, monitoring, etc.?
Please explain the types and distribution of ponds in the region. It seems that some ponds are used for swimming. Are they private or communal?
The paper is well written and generally clear. There are some minor grammatical errors e.g. unnecessary commas, redundant words such as: L 36 and L 436 ‘aged 1-4 years old’
Specific corrections
L 112 and L 386 mention Gram Panchayats, but this term is not explained until L 398
L 431 the term ‘meso-level’ might be unfamiliar to some readers – perhaps explain?
